# DeepSVG: A Hierarchical Generative Network for Vector Graphics Animation

**Alexandre Carlier**[1,2]     **Martin Danelljan**[2]     **Alexandre Alahi**[1]     **Radu Timofte**[2]

[1] Ecole Polytechnique Fédérale de Lausanne     [2] ETH Zurich

## Abstract

Scalable Vector Graphics (SVG) are ubiquitous in modern 2D interfaces due to their ability to scale to different resolutions. However, despite the success of deep learning-based models applied to rasterized images, the problem of vector graphics representation learning and generation remains largely unexplored. In this work, we propose a novel hierarchical generative network, called DeepSVG, for complex SVG icons generation and interpolation. Our architecture effectively disentangles high-level shapes from the low-level commands that encode the shape itself. The network directly predicts a set of shapes in a non-autoregressive fashion. We introduce the task of complex SVG icons generation by releasing a new large-scale dataset along with an open-source library for SVG manipulation. We demonstrate that our network learns to accurately reconstruct diverse vector graphics, and can serve as a powerful animation tool by performing interpolations and other latent space operations. Our code is available at `https://github.com/alexandre01/deepsvg`.

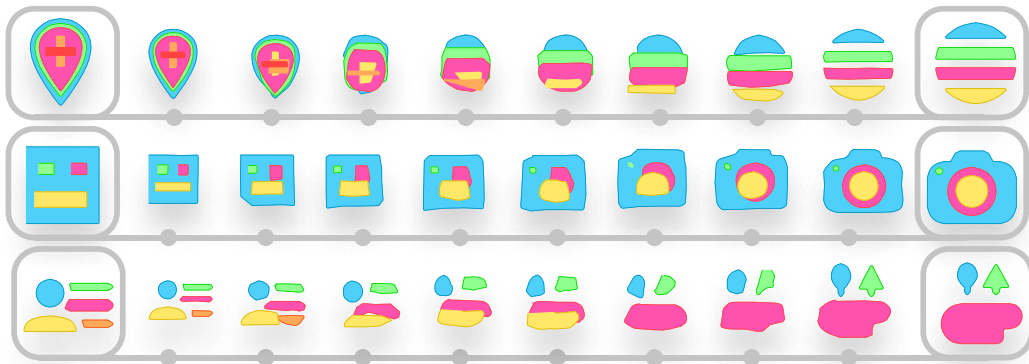

Figure 1: DeepSVG generates **vector graphics** by predicting draw commands, such as lines and Bézier curves. Our latent space allows meaningful animations between complex vector graphics icons.

## 1   Introduction

Despite recent success of rasterized image generation and content creation, little effort has been directed towards generation of vector graphics. Yet, vector images, often in the form of Scalable Vector Graphics [20] (SVG), have become a standard in digital graphics, publication-ready image assets, and web-animations. The main advantage over their rasterized counterpart is their scaling ability, making the same image file suitable for both tiny web-icons or billboard-scale graphics. Generative models for vector graphics could serve as powerful tools, allowing artists to generate, manipulate, and animate vector graphics, potentially enhancing their creativity and productivity.

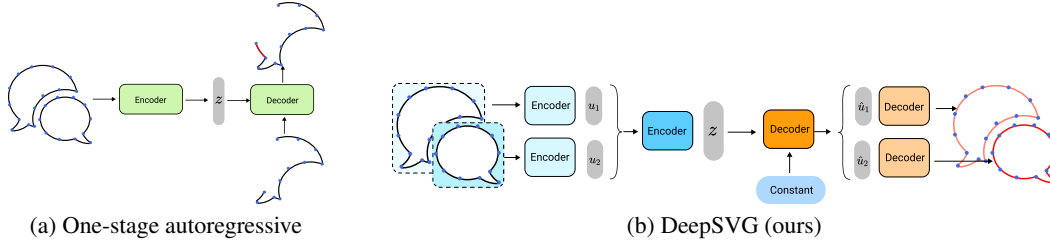

(a) One-stage autoregressive                   (b) DeepSVG (ours)

Figure 2: One-stage autoregressive autoencoder architectures [5, 11, 17] (a) take the entire draw commands as input and decode the latent vector one command at a time. Our approach (b) exploits the hierarchical nature of vector graphics in both the encoder and decoder, and decodes the draw commands with a single forward pass (non-autoregressively).

Raster images are most often represented as a rectangular grid of pixels containing a shade or color value. The recent success of deep learning on these images much owes to the effectiveness of convolutional neural networks (CNNs) [9], learning powerful representations by taking advantage of the inherent translational invariance. On the other hand, vector images are generally represented as lists of 2D shapes, each encoded as sequence of 2D points connected by parametric curves. While this brings the task of learning SVG representations closer to that of sequence generation, there are fundamental differences with other applications, such as Natural Language Processing. For instance, similar to the translation invariance in raster images, an SVG image experiences permutation invariance as the order of shapes in an SVG image is arbitrary. This brings important challenges in the design of both architectures and learning objectives.

We address the task of learning generative models of complex vector graphics. To this end, we propose a Hierarchical Transformer-based architecture that effectively disentangles high-level shapes from the low-level commands that encode the shape itself. Our encoder exploits the permutation invariance of its input by first encoding every shape separately, then producing the latent vector by reasoning about the relations between the encoded shapes. Our decoder mirrors this 2-stage approach by first predicting, in a single forward pass, a set of shape representations along with their associated attributes. These vectors are finally decoded into sequences of draw commands, which combined produce the output SVG image. A schematic overview of our architecture is given in Fig. 2.

**Contributions** Our contributions are three-fold: **1.** We propose DeepSVG, a hierarchical transformer-based generative model for vector graphics. Our model is capable of both encoding and predicting the draw commands that constitute an SVG image. **2.** We perform comprehensive experiments, demonstrating successful interpolation and manipulation of complex icons in vector-graphics format. Examples are presented in Fig. 1. **3.** We introduce a large-scale dataset of SVG icons along with a framework for deep learning-based SVG manipulation, in order to facilitate further research in this area. To the best of our knowledge, this is the first work to explore generative models of complex vector graphics, and to show successful interpolation and manipulation results for this task.

## 2 Related Work

Previous works [18, 13] for icon and logo generation mainly address rasterized image, by building on Generative Adversarial Networks [3]. Unlike raster graphics, vector graphics generation has not received extensive attention yet, and has been mostly limited to high-level shape synthesis [2] or *sketch* generation, using the 'Quick, Draw!' [19] dataset. SketchRNN [5] was the first Long Short Term Memory (LSTM) [6] based variational auto-encoder (VAE) [7] addressing the generation of sketches. More recently, Sketchformer [17] has shown that a Transformer-based architecture enables more stable interpolations between sketches, without tackling the generation task. One reason of this success is the ability of transformers [21] to more effectively represent long temporal dependencies.

SVG-VAE [11] was one of the first deep learning-based works that generate full vector graphics outputs, composed of straight lines and Bézier curves. However, it only tackles glyph icons, without global attributes, using an LSTM-based model. In contrast, our work considers the hierarchical nature of SVG images, crucial for representing and generating arbitrarily complex vector graphics. Fig. 2 compares previous one-stage autoregressive approaches [5, 11, 17] to our hierarchical architecture. Our work is also related to the very recent PolyGen [15] for generating 3D polygon meshes using sequential prediction vertices and faces using a Transformer-based architecture.

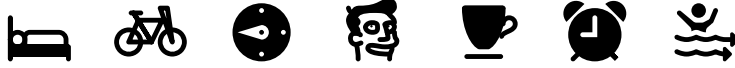

Figure 3: Samples from the *SVG-Icons8* dataset in vector graphics format. Although icons have similar scale and style, they have drastically diverse semantic meanings, shapes and number of paths.

## 3 DeepSVG

Here, we introduce our DeepSVG method. First, we propose a dataset of complex vector graphics and describe the SVG data representation in Sec. 3.1. We describe our learned embedding in Sec. 3.2. Finally, we present our architecture in Sec. 3.3 and training strategy in Sec. 3.4.

### 3.1 SVG Dataset and Representation

**SVG-Icons8 Dataset.** Existing vector graphics datasets either only contain straight lines [19] or are constrained to font generation [11]. These datasets therefore do not pose the challenges associated with the generation of complex vector graphics, addressed in this work. Thus, we first introduce a new dataset, called *SVG-Icons8*[1]. It is composed of SVG icons obtained from the `https://icons8.com` website. In the compilation of the dataset, we carefully considered the *consistency* and *diversity* of the collected icons. This was mainly performed by ensuring that the vector graphics have similar scale, colors and style, while capturing diverse real-world graphics allowing to learn meaningful and generalizable shape representations. In summary, our dataset consists of 100,000 high-quality icons in 56 different categories. Samples from the dataset are shown in Fig. 3. We believe that the *SVG-Icons8* dataset constitutes a challenging new benchmark for the growing task of vector graphics generation and representation learning.

**Vector Graphics and SVG.** In contrast to Raster graphics, where the content is represented by a rectangular grid of pixels, Vector graphics employs in essence mathematical formulas to encode different shapes. Importantly, this allows vector graphics to be scaled without any aliasing or loss in detail. Scalable Vector Graphics (SVG) is an XML-based format for vector graphics [20]. In its simplest form, an SVG image is built up hierarchically as a set of shapes, called *paths*. A path is itself defined as a sequence of specific draw-commands (see Tab. 1) that constitute a closed or open curve.

**Data structure.** In order to learn deep neural networks capable of encoding and predicting vector graphics, we first need a well defined and simple representation of the data. This is obtained by adopting the SVG format with the following simplifications. We employ the commands listed in Tab. 1. In fact, this does not significantly reduce the expressivity since other basic shapes can be converted into a sequence of Bézier curves and lines. We consider a Vector graphics image $V = \{P_1, \ldots, P_{N_P}\}$ to be a set of $N_P$ *paths* $P_i$. Each path is itself defined as a triplet $P_i =$

Table 1: List of the SVG draw-commands, along with their arguments and a visualization, used in this work. The start-position $(x_1, y_1)$ is implicitly defined as the end-position of the preceding command.

| Command | Arguments | Description | Visualization |
|---|---|---|---|
| `<SOS>` | $\varnothing$ | 'Start of SVG' token. | |
| M (MoveTo) | $x_2, y_2$ | Move the cursor to the end-point $(x_2, y_2)$ without drawing anything. | |
| L (LineTo) | $x_2, y_2$ | Draw a line to the point $(x_2, y_2)$. | |
| C (Cubic Bézier) | $q_{x1}, q_{y1}$ $q_{x2}, q_{y2}$ $x_2, y_2$ | Draw a cubic Bézier curve with control points $(q_{x1}, q_{y1})$, $(q_{x2}, q_{y2})$ and end-point $(x_2, y_2)$. | |
| z (ClosePath) | $\varnothing$ | Close the path by moving the cursor back to the path's starting position $(x_0, y_0)$. | |
| `<EOS>` | $\varnothing$ | 'End of SVG' token. | |

[1]Available at `https://github.com/alexandre01/deepsvg`.

$(S_i, f_i, v_i)$, where $v_i \in \{0, 1\}$ indicates the visibility of the path and $f_i \in \{0, 1, 2\}$ determines the fill property. Each $S_i = (C_i^1, \ldots, C_i^{N_C})$ contains a sequence of $N_C$ *commands* $C_i^j$. The command $C_i^j = (c_i^j, X_i^j)$ itself is defined by its type $c_i^j \in \{\texttt{<SOS>}, \texttt{m}, \texttt{l}, \texttt{c}, \texttt{z}, \texttt{<EOS>}\}$ and arguments, as listed in Tab. 1. To ensure efficient parallel processing, we use a fixed-length argument list $X_i^j = (q_{x_1,i}^j, q_{y_1,i}^j, q_{x_2,i}^j, q_{y_2,i}^j, x_{2,i}^j, y_{2,i}^j) \in \mathbb{R}^6$, where any unused argument is set to $-1$. Therefore, we also use a fixed number of paths $N_P$ and commands $N_C$ by simply padding with invisible elements in each case. Further details are given in appendix.

## 3.2 SVG Embedding

By the discrete nature of the data and in order to let the encoder reason between the different commands, every $C_i^j$ is projected to a common continuous embedding space of dimension $d_E$, similarly to the *de facto* approach used in Natural Language Processing [21]. This enables the encoder to perform operations across embedded vectors and learn complex dependencies between argument types, coordinate values and relative order of commands in the sequence. We formulate the embedding of the SVG command in a fashion similar to [1]. In particular, the command $C_i^j$ is embedded to a vector $e_i^j \in \mathbb{R}^{d_E}$ as the sum of three embeddings, $e_i^j = e_{\text{cmd},i}^j + e_{\text{coord},i}^j + e_{\text{ind},i}^j$. We describe each individual embedding next.

**Command embedding.** The command type (see Tab. 1) is converted to a vector of dimension $d_E$ using a learnable matrix $W_{\text{cmd}} \in \mathbb{R}^{d_E \times 6}$ as $e_{\text{cmd},i}^j = W_{\text{cmd}}\, \delta_{c_i^j} \in \mathbb{R}^{d_E}$, where $\delta_{c_i^j}$ designates the 6-dimensional one-hot vector containing a 1 at the command index $c_i^j$.

**Coordinate embedding.** Inspired by works such as PixelCNN [16] and PolyGen [15], which discretize continuous signals, we first quantize the input coordinates to 8-bits. We also include a case indicating that the coordinate argument is unused by the command, thus leading to an input dimension of $2^8 + 1 = 257$ for the embedding itself. Each coordinate is first embedded separately with the weight matrix $W_X \in \mathbb{R}^{d_E \times 257}$. The combined result of each coordinate is then projected to a $d_E$-dimensional vector using a linear layer $W_{\text{coord}} \in \mathbb{R}^{d_E \times 6 d_E}$,

$$e_{\text{coord},i}^j = W_{\text{coord}} \operatorname{vec}\left(W_X X_i^j\right), \quad X_i^j = \left[ q_{x_1,i}^j \; q_{y_1,i}^j \; q_{x_2,i}^j \; q_{y_2,i}^j \; x_{2,i}^j \; y_{2,i}^j \right] \in \mathbb{R}^{257 \times 6}. \quad (1)$$

Here, $\operatorname{vec}(\cdot)$ denotes the vectorization of a matrix.

**Index embedding.** Similar to [1], we finally use a learned index embedding[2] that indicates the index of the command in the given sequence using the weight $W_{\text{ind}} \in \mathbb{R}^{d_E \times N_S}$ as $e_{\text{ind},i}^j = W_{\text{ind}}\, \delta_j \in \mathbb{R}^{d_E}$, where $\delta_j$ is the one-hot vector of dimension $N_S$ filled with a 1 at index $j$.

## 3.3 Hierarchical Generative Network

In this section, we describe our Hierarchical Generative Network architecture for complex vector graphics interpolation and generation, called DeepSVG. A schematic representation of the model is shown in Fig. 4. Our network is a variational auto-encoder (VAE) [7], consisting of an encoder and a decoder network. Both networks are designed by considering the hierarchical representation of an SVG image, which consists of a set of paths, each path being a sequence of commands.

**Feed-forward prediction.** For every path, we propose to predict the $N_C$ commands $(\hat{c}_i^j, \hat{X}_i^j)$ in a purely feed-forward manner, as opposed to the autoregressive strategy used in previous works [5, 11], which learns a model predicting the next command conditioned on the history. Our generative model is thus factorized as,

$$p\left(\hat{V}|z, \theta\right) = \prod_{i=1}^{N_P} p(\hat{v}_i|z, \theta) p(\hat{f}_i|z, \theta) \prod_{j=1}^{N_C} p(\hat{c}_i^j|z, \theta) p(\hat{X}_i^j|z, \theta), \quad (2)$$

where $z$ is the latent vector and $p(\hat{X}_i^j|z, \theta)$ further factorizes into the individual arguments.

We found our approach to lead to significantly better reconstructions and smoother interpolations, as analyzed in Sec. 4. Intuitively, the feed-forward strategy allows the network to primarily rely on the

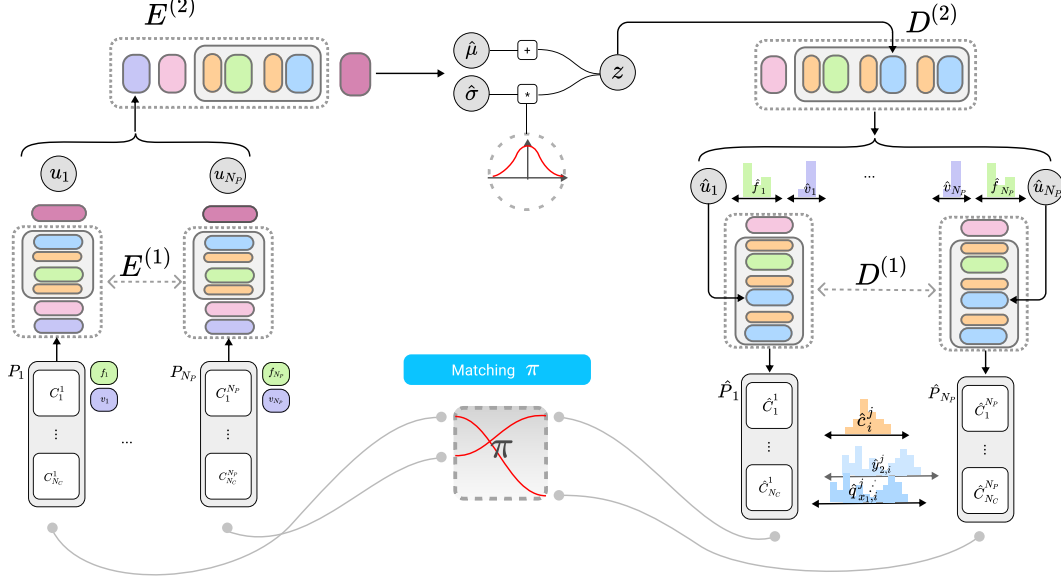

Figure 4: Our Hierarchical Generative Network, *DeepSVG*. Input paths $\{P_i\}_1^{N_P}$ are encoded separately using the path encoder $E^{(1)}$. The encoded vectors are then aggregated using the second encoder $E^{(2)}$, which produces the latent vector $z$. The decoder $D^{(2)}$ outputs the path representations along with their fill and visibility attributes $\{(\hat{u}_i, \hat{f}_i, \hat{v}_i)\}_1^{N_P}$. Finally $\{\hat{u}_i\}_1^{N_P}$ are decoded independently using the path decoder $D^{(1)}$, which outputs the actual draw commands and arguments.

latent encoding to reconstruct the input, without taking advantage of the additional information of previous commands and arguments. Importantly, a feed-forward model brings major advantages during training, since inference can be directly modeled during training. On the other hand, autoregressive methods [4, 21] condition on ground-truth to ensure efficient training through masking, while the inference stage conditions on the previously generated commands.

**Transformer block.** Inspired by the success of transformer-based architectures for a variety of tasks [17, 10, 1, 22], we also adopt it as the basic building block for our network. Both the Encoders and the Decoders are Transformer-based. Specifically, as in [17], we use $L = 4$ layers, with a feed-forward dimension of 512 and $d_E = 256$.

**Encoder.** To keep the permutation invariance property of the paths set $\{P_i\}_1^{N_P}$, we first encode every path $P_i$ independently using *path encoder* $E^{(1)}$. More specifically, $E^{(1)}$ takes the embeddings $(e_i^j)_{j=1}^{N_C}$ as input and outputs vectors $(e'^j_i)_{j=1}^{N_C}$ of same dimension. To retrieve the single $d_E$-dimensional path encoding $u_i$, we average-pool the output vectors along the sequential dimension. The $N_P$ path encodings $\{u_i\}_1^{N_P}$ are then input in encoder $E^{(2)}$ which, after pooling along the set-dimension, outputs the parameters of a Gaussian distribution $\hat{\mu}$ and $\hat{\sigma}$. Note how the index embedding in vector $e_i^j$ enables $E^{(1)}$ to reason about the sequential nature of its input while $E^{(2)}$ maintains the permutation invariance of the input paths. The latent vector is finally obtained using the reparametrization trick [7] as $z = \hat{\mu} + \hat{\sigma} \cdot \epsilon$, where $\epsilon \sim \mathcal{N}(0, I)$.

**Decoder.** The decoder mirrors the two-stage construction of the encoder. $D^{(2)}$ inputs the latent vector $z$ repeatedly, at each transformer block, and predicts a representation of each shape in the image. Unlike the corresponding encoder stage, permutation invariance is not a desired property for $D^{(2)}$, since its purpose is to generate the shapes in the image. We achieve this by using a learned index embedding as input to the decoder. The embeddings are thus distinct for each path, breaking the symmetry during generation. The decoder is followed by a Multilayer Perceptron (MLP) that outputs, for each index $1 \leq i \leq N_P$, the predicted path encoding $\hat{u}_i$, filling $\hat{f}_i$ and visibility $\hat{v}_i$ attributes. Symmetrically to the encoder, the vectors $\{\hat{u}_i\}_1^{N_P}$ are decoded by $D^{(1)}$ into the final output path representations $\{(\hat{C}_i^1, \cdots, \hat{C}_i^{N_C})\}_1^{N_P}$. As for $D^{(2)}$, we use learned constant embeddings as input and a MLP to predict the command and argument logits. Detailed descriptions about the architectures are given in the appendix.

## 3.4 Training Objective

Next, we present the training loss used by our DeepSVG. We first define the loss between a predicted path $(\hat{S}_{\hat{i}}, \hat{f}_{\hat{i}}, \hat{v}_{\hat{i}})$ and a ground-truth path $(S_i, f_i, v_i)$ as,

$$L_{\hat{i},i}(\theta) = w_{\text{vis}}\ell(v_i, \hat{v}_i) + v_i \cdot \left( w_{\text{fill}}\ell(f_i, \hat{f}_i) + \sum_{j=1}^{N_C} \left( w_{\text{cmd}}\ell(c_{\hat{i}}^j, \hat{c}_i^j) + w_{\text{args}}\, l_{\text{args},\hat{i},i}^j \right) \right). \quad (3)$$

Here, $\ell$ denotes the Cross-Entropy loss. The impact of each term is controlled by its weight $w$. The losses for filling, commands and arguments are masked when the groundtruth path is not visible. The loss $l_{\text{args},\hat{i},i}^j$ over the argument prediction is defined as,

$$l_{\text{args},\hat{i},i}^j = \mathbf{1}_{c_i^j \in \{\text{m,l,c}\}} \left( \ell(x_{2,\hat{i}}^j, \hat{x}_{2,i}^j) + \ell(y_{2,\hat{i}}^j, \hat{y}_{2,i}^j) \right) + \mathbf{1}_{c_i^j = \text{c}} \sum_{k \in \{1,2\}} \ell(q_{x_k,\hat{i}}^j, \hat{q}_{x_k,i}^j) + \ell(q_{y_k,\hat{i}}^j, \hat{q}_{y_k,i}^j). \quad (4)$$

Having formulated the loss for a single path, the next question regards how to this can be used to achieve a loss on the entire prediction. However, recall that the collection of paths in a vector image has no natural ordering, raising the question of how to assign ground-truth paths to each prediction. Formally, a ground-truth *assignment* $\pi$ is a permutation $\pi \in \mathcal{S}_{N_P}$, mapping the path index of the prediction $\hat{i}$ to the corresponding ground-truth path index $i = \pi(\hat{i})$. We discuss two alternatives for solving the ground-truth assignment problem.

**Ordered assignment.** One strategy is to define the assignment $\pi$ by sorting the ground-truth paths according to some specific criterion. This induces an ordering $\pi_{\text{ord}}$, which the network learns to reproduce. We found defining the ground-truth assignment using the lexicographic order of the starting location of the paths to yield good results. Given any sorting criterion, the loss is defined as,

$$L(\theta) = w_{\text{KL}}\text{KL}\left(p_\theta(z) \| \mathcal{N}(0, I)\right) + \sum_{\hat{i}=1}^{N_P} L_{\hat{i}, \pi_{\text{ord}}(\hat{i})}(\theta), \quad (5)$$

where the first term corresponds to the latent space prior induced by the VAE learning.

**Hungarian assignment.** We also investigate a strategy that does not require defining a sorting criterion. For each prediction, we instead find the best possible assignment $\pi$ in terms of loss,

$$L(\theta) = w_{\text{KL}}\text{KL}\left(p_\theta(z) \| \mathcal{N}(0, I)\right) + \min_{\pi \in \mathcal{S}_{N_P}} \sum_{\hat{i}=1}^{N_P} L_{\hat{i}, \pi(\hat{i})}(\theta). \quad (6)$$

The best permutation is found through the Hungarian algorithm [8, 14].

**Training details.** We use the AdamW [12] optimizer with initial learning rate $10^{-4}$, reduced by a factor of 0.9 every 5 epochs and a linear warmup period of 500 initial steps. We use a dropout rate of 0.1 in all transformer layers and gradient clipping of 1.0. We train our networks for 100 epochs with a total batch-size of 120 on two 1080Ti GPUs, which takes about one day.

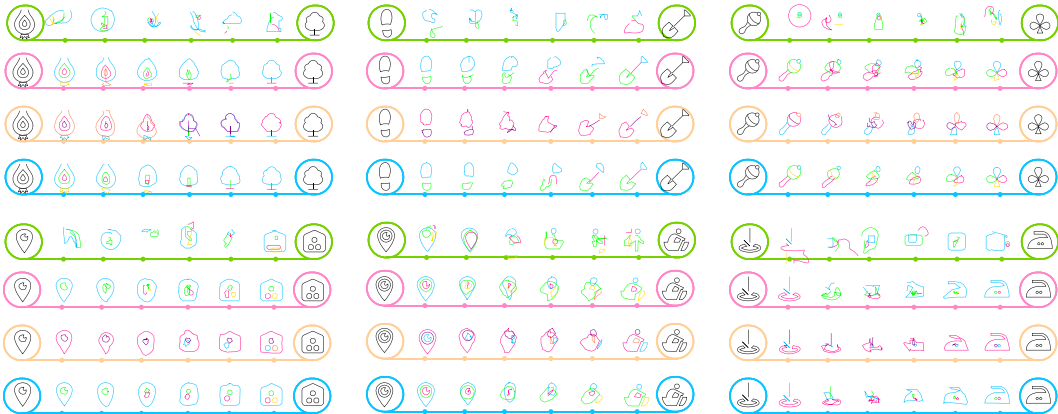

Figure 5: Comparison of interpolations between one-stage autoregressive (top row, in green), one-stage feed-forward (2$^{\text{nd}}$ row, in pink), ours – Hungarian (3$^{\text{rd}}$ row, in orange) and ours – ordered (bottom row, in blue). Ordered generally leads to the smoothest interpolations. The last two examples show interpolations where Hungarian yields visually more meaningful shape transitions. For a better visualization of these transitions, paths are colored according to their index (or order for one-stage architectures).

Table 2: Ablation study of our DeepSVG model showing results of the human study (1st rank % and average rank), and quantitative measurements (RE and IS) on train/test set.

| | Feed-forward | Hierarchical | Matching | 1st rank % ↑ | Average rank ↑ | RE ↓ | IS ↓ |
|---|---|---|---|---|---|---|---|
| One-stage autoregressive | | | | 9.7 | 3.26 | 0.102 / 0.170 | 0.25 / 0.36 |
| One-stage feed-forward | ✓ | | | 19.5 | 2.40 | **0.007** / 0.014 | 0.12 / 0.17 |
| Ours – Hungarian | ✓ | ✓ | Hungarian | 25.8 | 2.29 | 0.011 / 0.017 | 0.09 / 0.14 |
| **Ours – Ordered** | ✓ | ✓ | **Ordered** | **44.8** | **1.99** | **0.007 / 0.012** | **0.08 / 0.12** |

# 4 Experiments

We validate the performance of our DeepSVG method on the introduced SVG-Icons8 dataset. We also demonstrate results for glyph generation on the SVG-Fonts [11] dataset. Further experiments are presented in the supplementary material.

## 4.1 Ablation study

In order to ablate our model, we first evaluate an autoregressive one-stage architecture by concatenating the set of (unpadded) input sequences, sorted using the Ordered criterion 3.4. The number of paths therefore becomes $N_P = 1$ and only Encoder $E^{(1)}$ and Decoder $D^{(1)}$ are used; filling is ignored in that case. We analyze the effect of feed-forward prediction, and then our hierarchical DeepSVG architecture, using either the Ordered or Hungarian assignment loss 3.4.

**Human study.** We conduct a human study by randomly selecting 100 pairs of SVG icons, and showing the interpolations generated by the four models to 10 human participants, which rank them best (1) to worst (4). In Tab. 2 we present the results of this study by reporting the percentage of 1st rank votes, as well as the average rank for each model. We also show qualitative results in Fig. 5, here ignoring the filling attribute since it is not supported by one-stage architectures.

**Quantitative measures.** To further validate the performance of DeepSVG, we conduct a quantitative evaluation of all the methods. We therefore here propose two vector image generation metrics. We first define the Chamfer distance between two SVGs: $d_{\text{Chfr}}(V, \hat{V}) = \frac{1}{N_P} \sum_{i=1}^{N_P} \min_j \int_t \min_\tau \|P_i(t) - \hat{P}_j(\tau)\|_2 dt$, where $P_i \in V$ is a path as defined in 3.1. The *Reconstruction Error* (RE) is $d_{\text{Chfr}}(V, \hat{V})$ where $V$ and $\hat{V}$ are the ground-truth and reconstruction. The *Interpolation Smoothness* (IS) is defined as $\sum_{k=1}^{M} d_{\text{Chfr}}(V^{\alpha_{k-1}}, V^{\alpha_k})$, where $M$ is the number of frames, $\alpha_k = k/M$ and $V^\alpha$ is the predicted SVG interpolation parametrized by $\alpha \in [0, 1]$. Results on train and test sets are shown in Tab. 2.

Compared to the autoregressive baseline, the use of feed-forward prediction brings substantial improvement in reconstruction error and interpolation quality, as also confirmed by the qualitative results. In the human study, our hierarchical architecture with ordered assignment yields superior results. Although providing notably better reconstruction quality, this version provides much more stable and meaningful interpolations compared to the other approaches. The Hungarian assignment achieves notably worse results compared to ordered assignment in average. Note that the latter is more related to the loss employed for the one-stage baselines, although there acting on a command level. We hypothesize that the introduction of a sensible ordering during training helps the decoder learning by providing an explicit prior, which better breaks symmetries and reduces competition between the predicted paths. Fig. 6 further shows how the latent SVG representation translates to meaningful decodings by performing interpolation between 4 SVG icons.

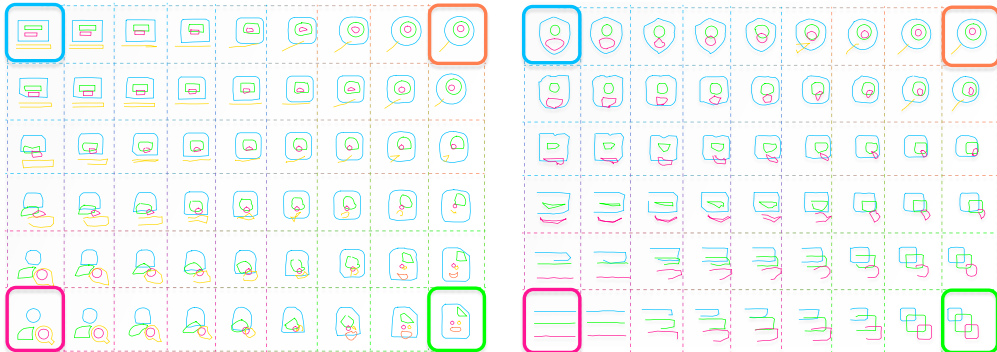

Figure 6: Interpolation between multiple icons in the latent space of DeepSVG – Ordered.

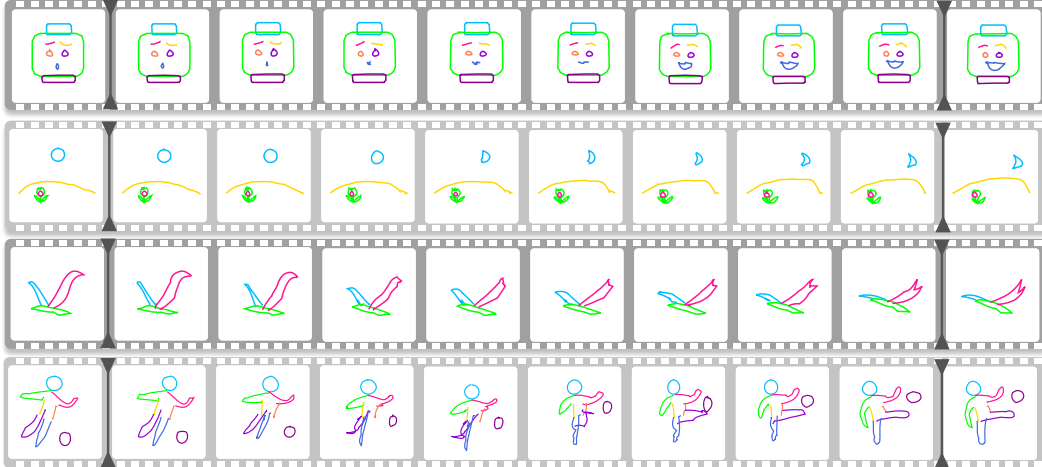

Figure 7: Animating SVG scenes by interpolation. Leftmost and rightmost frames are drawn by a user, while images in between are interpolations. DeepSVG smoothly interpolates between challenging path deformations while accurately reconstructing the 1$^{\text{st}}$ and last frames. A failure case is shown in the last row where the deformation of the player's right leg is not smoothly interpolated.

## 4.2 Animation by interpolation

As visually demonstrated in the previous subsection, we observe significantly better reconstruction capability of our model than previous works. This property is crucial for real-world applications involving SVGs since users should be able to perform various operations on vector graphics while keeping their original drawings unchanged. With this requirement in mind, we examine if DeepSVG can be used to animate SVGs. We investigate *interpolation* as one approach to perform it, in the setting where a user provides two keyframes and wants to generate frames inbetween using shape morphing. This process can be repeated iteratively – adding a hand-drawn keyframe at every step – until a satisfying result is achieved. Fig. 7 shows the results of challenging scenes, after finetuning the model on both keyframes for about 1,000 steps. Notice how DeepSVG handles well both translations and deformations.

## 4.3 Latent space algebra

Given DeepSVG's smooth latent space and accurate reconstruction ability, we next ask if latent directions may enable to manipulate SVGs globally in a semantically meaningful way. We present two experiments in Fig. 8. In both cases, we note $\Delta$ the difference between encodings of two similar SVGs differing by some visual semantics. We show how this latent direction can be added or subtracted to the latent vector $z$ of arbitrary SVG icons. More experiments are presented in the appendix. In particular, we examine whether DeepSVG's hierarchical construction enables similar operations to be performed on single paths instead of globally.

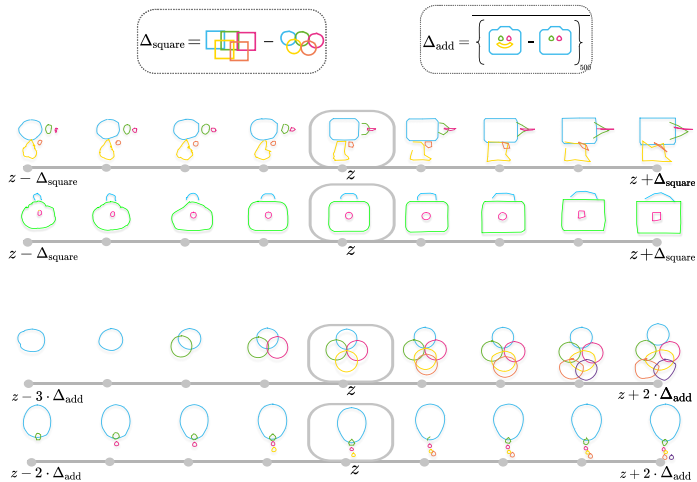

Figure 8: Global operations on SVG representations using latent directions. Subtracting/adding $\Delta_{\text{square}}$ makes an icon look more round/rectangular, while $\Delta_{\text{add}}$ adds or removes paths. $\Delta_{\text{add}}$ is obtained by removing the last path of an icon, and averaging the difference over 500 random icons.

### 4.4 Font generation

Our experiments have demonstrated so far reconstruction, interpolation and manipulation of vector graphics. In this section, we further show the generative capability of our method, by decoding random vectors sampled from the latent space. We train our models on the SVG-Fonts dataset, for the task of class-conditioned glyph generation. DeepSVG is extended by adding label embeddings at every layer of each Transformer block. We compare the generative capability of our final model with the same baselines as in Sec. 4.1. In addition, we show random samples from SVG-VAE [11]. Results are shown in Fig. 9. Notice how the non-autoregressive settings generate consistently visually more precise font characters, without having to pick the best example from a larger set nor using any post-processing. We also note that due to the simplicity of the SVG-Font dataset, no significant visual improvement from our hierarchical architecture can be observed here. More details on the architecture and results are shown in the appendix.

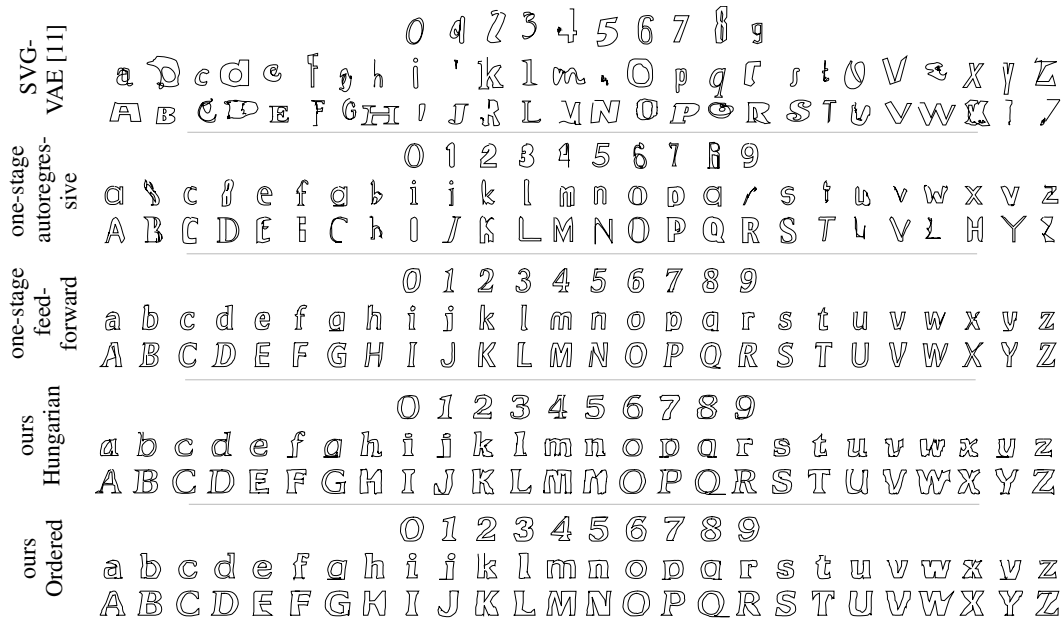

Figure 9: Comparison of samples for font generation. We use the same latent vector $z$, sampled from a Gaussian distribution with standard deviation $\sigma = 0.5$, and condition on each class label without careful selection of generated samples nor post-processing.

## 5 Conclusion

We have demonstrated how our hierarchical network can successfully perform SVG icons interpolations and manipulation. We hope that our architecture will serve as a strong baseline for future research in this, to date, little-explored field. Interesting applications of our architecture include image vectorisation, style transfer (Sec. 4.3), classification, animation, or the more general task of XML generation by extending the two-level hierarchy used in this work. Furthermore, while DeepSVG was designed specifically for the natural representation of SVGs, our architecture can be used for any task involving data represented as a *set* of *sequences*. We therefore believe it can be used, with minimal modifications, in a wide variety of tasks, including multi-instrument audio generation, multi-human motion trajectory generation, etc.

## Broader Impact

DeepSVG can be used as animation tool by performing interpolations and other latent space operations on user-drawn SVGs. Similarly to recent advances in rasterized content creation, we believe this work will serve as a potential way for creators and digital artists to enhance their creativity and productivity.

## Acknowledgments and Disclosure of Funding

This work was partly supported by the ETH Zürich Fund (OK), a Huawei Technologies Oy (Finland) project, an Amazon AWS grant, and an Nvidia hardware grant.

## Footnotes

[2]Known as *positional embedding* in the Natural Language Processing literature [21].

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
