[Supplementary Material]

# Appendix

In this appendix, we first present a visualization of the data structure used for SVGs in Sec. A. We provide detailed instructions used to preprocess our data in Sec.B. Additional details on training and architectures are given in Sec. C and Sec. D. Sec. E goes through the procedure to predict filling along with SVG paths. Finally, additional results for font generation, icons generation, latent space algebra, animations and interpolations are presented in sections F, G, H, I and J respectively.

## A    SVG Representation visualization

For a visual depiction of the data structure described in Sec. 3.1, we present in Fig. 1 an example of SVG image along with its tensor representation. The SVG image consists of 2 paths, $P_1$ and $P_2$. The former, $P_1$ starts with a move `m` command from the top left corner. The arc is constructed from two Cubic Bézier curve `c` commands. This is followed by a line `l` and close path `z` command. The `<EOS>` command indicates the end of path $P_1$. $P_2$ is constructed in a similar fashion using only a single Cubic Bézier curve.

Figure 1: Example of SVG representation. Left: Input SVG image. Right: Corresponding tensor representations with 2 paths and 7 commands ($N_P = 2, N_C = 7$). Commands in the image and the corresponding tensor are color-coded for a better visualization. The arguments are listed in the order $q_{x1}$, $q_{y1}$, $q_{x2}$, $q_{y2}$, $x_2$ and $y_2$. Best viewed in color.

## B    SVG Preprocessing

In Sec. 3.1, we consider that SVG images are given as a set of paths, restricted to the 6 commands described in Tab. 1. As mentioned, this does not reduce the expressivity of vector graphics since other basic shapes and commands can be converted to that format. We describe next the details of these conversions.

**Path commands conversion.** Lower-case letters in SVG path commands are used to specify that their corresponding arguments are *relative* to the preceding command's end-position, as opposed to *absolute* for upper-case letters. We start by converting all commands to absolute. Other available commands (`H`: HorizonalLineTo, `V`: VerticalLineTo, `S`: SmoothBezier, `Q`: QuadraticBezier, `T`: SmoothQuadraticBezier) can be trivially converted to the commands subset of Tab. 1. The only missing command that needs further consideration is the elliptical-arc command `A`, described below.

**Elliptical arc conversion.** As illustrated in Fig. 2, command `A` $r_x$, $r_y$ $\varphi$ $f_A$ $f_S$ $x_2$, $y_2$ draws an elliptical arc with radii $r_x$ and $r_y$ (semi-major and semi-minor axes), rotated by angle $\varphi$ to the $x$-axis, and end-point $(x_2, y_2)$. The bit-flags $f_A$ and $f_S$ are used to uniquely determine which one of the four possible arcs is chosen: large-arc-flag $f_A$ is set to 1 if the arc spanning more than 180° is chosen, 0 otherwise; and sweep-flag $f_s$ is set to 0 if the arc is oriented clockwise, 1 otherwise. We argue that this parametrization, while being intuitive from a user-perspective, adds unnecessary complexity to the commands argument space described in Sec.3.1 and the bit-flags make shapes non-continuous w.r.t. their arguments, which would result in less smooth animations. We therefore convert `A` commands to multiple Cubic Bézier curves. We first start by converting the endpoint parametrization $(x_1, y_1), (x_2.y_2)$ to a center parametrization $(c_x, c_y)$. The center of the ellipse is

Figure 2: Elliptical Arc command visualization. The command is parametrized with arguments: $r_x$, $r_y$, $\varphi$, $f_A$, $f_S$, $x_2$ and $y_2$.

computed using:

$$\begin{pmatrix} c_x \\ c_y \end{pmatrix} = \begin{pmatrix} \cos\varphi & -\sin\varphi \\ \sin\varphi & \cos\varphi \end{pmatrix} \begin{pmatrix} c_x' \\ c_y' \end{pmatrix} + \begin{pmatrix} \frac{x_1+x_2}{2} \\ \frac{y_1+y_2}{2} \end{pmatrix} \tag{1}$$

where,

$$\begin{pmatrix} c_x' \\ c_y' \end{pmatrix} = \pm\sqrt{\frac{r_x^2 r_y^2 - r_x^2 (y_1')^2 - r_y^2 (x_1')^2}{r_x^2 (y_1')^2 + r_y^2 (x_1')^2}} \begin{pmatrix} \frac{r_x y_1'}{r_y} \\ -\frac{r_y x_1'}{r_x} \end{pmatrix} \tag{2}$$

$$\begin{pmatrix} x_1' \\ y_1' \end{pmatrix} = \begin{pmatrix} \cos\varphi & \sin\varphi \\ -\sin\varphi & \cos\varphi \end{pmatrix} \begin{pmatrix} \frac{x_1-x_2}{2} \\ \frac{y_1-y_2}{2} \end{pmatrix} \tag{3}$$

We then determine the *start angle* $\theta_1$ and *angle range* $\Delta\theta$ which are given by computing:

$$\theta_1 = \angle\left( \begin{pmatrix} 1 \\ 0 \end{pmatrix}, \begin{pmatrix} \frac{x_1'-c_x'}{r_x} \\ \frac{y_1'-c_y'}{r_y} \end{pmatrix} \right) \tag{4}$$

$$\Delta\theta = \angle\left( \begin{pmatrix} \frac{x_1'-c_x'}{r_x} \\ \frac{y_1'-c_y'}{r_y} \end{pmatrix}, \begin{pmatrix} \frac{-x_1'-c_x'}{r_x} \\ \frac{-y_1'-c_y'}{r_y} \end{pmatrix} \right) \tag{5}$$

Using $(c_x, c_y)$, $\theta_1$ and $\Delta\theta$, we obtain the parametric elliptical arc equation as follows (for $\theta$ ranging from $\theta_1$ to $\theta_1 + \Delta\theta$):

$$E(\theta) = \begin{pmatrix} c_x + r_x \cos\varphi \cos\theta - r_y \sin\varphi \sin\theta \\ c_y + r_x \sin\varphi \cos\theta - r_y \cos\varphi \sin\theta \end{pmatrix} \tag{6}$$

and the derivative of the parametric curve is:

$$E'(\theta) = \begin{pmatrix} -r_x \cos\varphi \sin\theta - r_y \sin\varphi \cos\theta \\ -r_x \sin\varphi \sin\theta - r_y \cos\varphi \cos\theta \end{pmatrix} \tag{7}$$

Given both equations, [5] shows that the section of elliptical arc between angles $\theta_1$ and $\theta_2$ can be approximated by a cubic Bézier curve whose control points are computed as follows:

$$\begin{cases} P_1 &= E(\theta_1) \\ P_2 &= E(\theta_2) \\ Q_1 &= P_1 + \alpha E'(\theta_1) \\ Q_2 &= P_2 - \alpha E'(\theta_2) \end{cases} \tag{8}$$

where $\alpha = \sin\theta_2 - \theta_1 \frac{\sqrt{4+3\tan^2 \frac{\theta_2-\theta_1}{2}}-1}{3}$.

**Basic shape conversion.** In addition to paths, SVG images can be built using 6 basic shapes: rectangles, lines, polylines, polygons, circles and ellipses. The first four can be converted to paths

Table 1: Examples of conversion from basic shapes (rectangle, circle, ellipse, line, polyline and polygon) to paths.

| Basic Shape | | Path equivalent | |
|---|---|---|---|
| | `<rect x="0" y="0"`<br>`      width="1" height="1" />` | `<path d="M0,0 L1,0 L1,1`<br>`          L0,1 L0,0 z" />` | |
| | `<circle cx="1" cy="1"`<br>`         r="1" />,`<br>`<ellipse cx="1" cy="1"`<br>`         rx="1" ry="1" />` | `<path d="M1,0 A1,1 0 0 1 2,1`<br>`          A1,1 0 0 1 1,2`<br>`          A1,1 0 0 1 0,1`<br>`          A1,1 0 0 1 1,0 z" />` | |
| | `<line x1="0" x2="1" y1="0"`<br>`       y2="1" />` | `<path d="M0,0 L1,1" />` | |
| | `<polyline points="0, 0 1, 0 1,`<br>`           1" />` | `<path d="M0,0 L1,0 L1,1" />` | |
| | `<polgon points="0, 0 1, 0 1, 1"`<br>`        />` | `<path d="M0,0 L1,0 L1,1 z" />` | |

using Line commands, while the latter two are transformed to a path using four Elliptical Arc commands, which themselves are converted to Bézier curves using the previous section. Table 1 below shows examples of these conversions.

**Path simplification.** Similarly to Sketch-RNN [3], we preprocess our dataset in order to simplify the network's task of representation learning. However, unlike the latter work, our input consists of both straight lines and parametric curves. Ideally, if shapes were completely smooth, one could reparametrize points on a curve so that they are placed equidistantly from one another. In practice though, SVG shapes contain sharp angles, at which location points should remain unchanged. We therefore first split paths at points that form a sharp angle (e.g. where the angle between the incoming and outgoing tangents is less than some threshold $\eta = 150°$). We then apply either the Ramer-Douglas-Peucker [2] algorithm to simplify line segments or the Philip J. Schneider algorithm [8] for segments of cubic Bézier curves. Finally, we divide the resulting lines and Bézier curves in multiple subsegments when their lengths is larger than some distance $\Delta = 5$. Examples of SVG simplifications are shown in Fig. 3. Notice how our algorithm both adds points when curve segments are too long or reduces the amount of points when the curve resolution is too high.

Figure 3: Examples of SVG simplifications. Top: original SVGs as downloaded from the `https://icons8.com` website. Bottom: Same icons after path simplification.

**SVG normalization.** All SVGs are scaled to a normalized *viewbox* of size $256 \times 256$, and paths are *canonicalized*, meaning that a shape's starting position is chosen to be the topmost leftmost point, and commands are oriented clockwise.

## C   Additional Training details

We augment every SVG of the dataset using 20 random augmentations with the simple transformations described as follows.

**Scaling.** We scale the SVG by a random factor $s$ in the interval $[0.8, 1.2]$.

**Translation.** We translate the SVG by a random translation vector $t$ where $t_x$ and $t_y$ are sampled independently in the interval $[-2.5, 2.5]$.

We believe further robustness in shape representation learning and interpolation stability can be obtained by simply implementing more complex data augmentation strategies.

## D   Architectural details

Fig. 4 presents an overview illustration of our Hierarchical autoencoder architecture. In Fig. 4, we here show a more detailed view of the four main components of DeepSVG, i.e. the two encoders $E^{(1)}$, $E^{(2)}$ and decoders $D^{(2)}$, $D^{(1)}$. Similarly to [6], we use the improved Transformer variant described in [1, 7] as building block in all our components. $E^{(1)}$ and $E^{(2)}$ employ a temporal pooling module to retrieve a single $d_E$-dimensional vector from the $N_C$ and $N_P$ outputs respectively. $D^{(2)}$ and $D^{(1)}$ use learned embeddings as input in order to generate all predictions in a single forward-pass (non-autoregressively) and break the symmetry. The decoders are conditioned on latent vector $z$ or path representation $u_i$ by applying a linear transformation and adding it to the intermediate transformer representation in every block.

Figure 4: Detailed view of architectures of $E^{(1)}$, $E^{(2)}$, $D^{(2)}$ and $D^{(1)}$.

# E   Filling procedure visualization

Thanks to its hierarchical construction, DeepSVG can predict any number of global path-level attributes, which could be e.g. color, dash size, stroke-width or opacity. As a first step towards a network modeling all path attributes supported by the SVG format, we demonstrate support for filling. When using the default `non-zero` fill-rule in the SVG specification, a point in an SVG path is considered inside or outside the path based on the draw orientations (clockwise or counter-clockwise) of the shapes surrounding it. In particular, the *insideness* of a point in the shape is determined by drawing a ray from that point to infinity in any direction, and then examining the places where a segment of the shape crosses the ray. Starting with a count of zero, add one each time a path segment crosses the ray from left to right and subtract one each time a path segment crosses the ray from right to left. We argue that this parametrization is not optimal for neural networks to encode filling/erasing. Therefore, we simply let the network output a *fill-attribute* that can take one of three values: *outline*, *fill* or *erase*. This attribute is trained in a supervised way along with the other losses and is then used to export the actual SVG file. In particular, overlapping *fill* and *erase* shapes are grouped together in a same path and oriented in a clockwise/counterclockwise fashion respectively, while *outlined* shapes remain unchanged.

Figure 5: Examples of icon interpolations when using the fill attribute, predicted by the global decoder $D^{(1)}$. Look at how shapes' filling generally changes at the middle of interpolation, while being deformed in a smooth way.

# F  Font generation

In this section, we provide details and additional results for font generation, presented in Sec. 4.4.

**Experimental setup.** We train our models on the SVG-Fonts dataset [4] for 5 epochs using the same training hyper-parameters as described in Sec. 3.4, reducing the learning rate by a factor 0.9 every quarter epoch. Furthermore, all encoder and decoder Transformer blocks are extended to be class-conditioned. Similarly to how latent vector $z$ is fed into $D^{(2)}$, we add the learned label embedding to the intermediate transformer representation, after liner transformation. This is done in $E^{(1)}$, $E^{(2)}$, $D^{(2)}$ and $D^{(1)}$ and applies for both our final model and the one-stage baselines.

**Additional results.** To validate that our model generates diverse font samples, we also present in Fig. 6 different samples for every glyph. Note how the latent vector $z$ is decoded into a style-consistent set of font characters. Diversity here includes different levels of boldness and more or less italic glyphs.

Figure 6: Font samples from DeepSVG – Ordered, generated from 7 different latent vectors. As observed in SVG-VAE [4], we notice a style consistency across the generated glyphs for a same latent vector. For instance, note how columns 5 & 6 correspond to an italic font style, column 4 to an extra thin one, and column 2 to a bolder one.

# G  Random samples of icons

In this section, we show random samples of icons by our model. Fig. 7 presents a set of icons generated by DeepSVG, obtained by sampling random latent vectors $z$. These results show diverse icons that look visually reasonable. Note that the problem of *generic icon* generation is much more challenging than font generation. Results are promising, but much scope for improvement remains.

Figure 7: Random samples of icons.

# H  Additional results on latent space algebra

As mentioned in Sec.4.3, operations on vectors in the latent space lead to semantically meaningful SVG manipulations. By the hierarchical nature of our architecture, we here demonstrate that such operations can also be performed at the *path*-level, using path encodings $(\hat{u}_i)_1^{N_P}$. In Fig. 8 we consider the difference $\Delta$ between path encodings of similar shapes, that differ by a horizontal or vertical translation. Adding or removing $\Delta$ from a path encoding in arbitrary SVG images applies the same translation to it.

Figure 8: Vector operations at the path-level. $\hat{u}_1$ corresponds to the path encoding of the blue shape, while $\hat{u}_2$ corresponds to the shape in green.

# I  Additional animations by interpolation

We here show three additional animations, generated by DeepSVG from two user-created drawings. DeepSVG handles well deformation, scaling and rotation of shapes, see Fig. 9.

Figure 9: Additional animation examples.

# J  Additional interpolations

Finally, we present additional interpolation results in Fig. 10 using our DeepSVG – ordered model, showing successful interpolations between challenging pairs of icons, along with some failure cases.

Figure 10: Additional interpolations of DeepSVG – ordered. The last two rows show examples of challenging icons, where interpolations appear visually less smooth.