[Reviews · NeurIPS 2020]

Review 1

Summary and Contributions: The paper builds an auto-encoding deep neural network to process SVG programs. By modelling the programs as a set of tokens using a transformer-based architecture, the authors propose a variational auto-encoder that is able to efficiently process simple SVG drawings. Furthermore, the authors propose a novel dataset to train such models and (hopefully) plan to release it along with the work. I believe the contribution is highly relevant to NeurIPS and deserves clear acceptance.

Strengths: The considered task is significantly novel with many applications such as image vectorization and generation of animations, and the presented model is also novel, being in essence a combination of the well-known building blocks, which suggests their stability. The authors conduct an extensive experimental validation building on the novel SVG dataset. The interpolation, generative, and other qualities of the model are clearly demonstrated.

Weaknesses: 1. The model is only able to process simple SVG drawings as of now, which indicates its limitation but does not diminish the novelty. 2. The paper is somewhat difficult to read (see below).

Correctness: The experimental methodology is overall valid but a few things need clarification: 1. Specifically, would SVG-VAE qualify for a baseline? If not, why? 2. What is named as "baseline" in the experimental session? 3. How is the baseline different from “One-stage feed-forward”?

Clarity: Overall the paper is challenging to read, however with a few areas that represent particular difficulty. Specifically, Page 5 is extremely difficult to comprehend. I suggest moving Lines 159—162 describing Transformer modules to before L139, describing Encoders.

Relation to Prior Work: Related work seemingly needs to include https://papers.nips.cc/paper/7845-learning-to-infer-graphics-programs-from-hand-drawn-images.pdf along with a short discussion of the relation between the two. Other than that, I have found to gaps in the related literature.

Reproducibility: Yes

Additional Feedback:


Review 2

Summary and Contributions: The paper discusses a method for modeling simple vector drawings such as icons or fonts. Differently to previous works the method is not autoregressive, in which the next step is predicted after previous steps were generated. In the current work, different paths are considered separately and then overlaid together with in the decoder. To model vector graphics the model uses a subset of SVG language with a command of fixed size. The commands and arguments are first encoded using encoding matrices. Then the transformer-based encoder is used. The latent is modeled by a gaussian distribution via VAE.

Strengths: The paper is an interesting piece of work. I agree, modeling vector is an under explored problem. 1. The paper proposed an interesting method of modeling vector graphics, based on a small yet flexible set of svg commands. The paper addressed several problems of how to deal with such entity in machine learning. 2. The paper proposed a dataset of SVG icons which might be interesting for research community. 3. It is also exciting to see that there is some latent space algebra available.

Weaknesses: When reading the paper, I've got the impression that the paper is not finished with couple of key experiments missing. Some parts of the paper lack motivation. Terminology is sometimes unclear and ambiguous. 1. Terminology. The paper uses terms "animation", "generative", "interpolation". See contribution 1 in L40-42. While the paper reported some interpolation experiments, I haven't found any animation or generation experiments. It seems the authors equate interpolation and animation (Section 4.2) which is not correct. I consider animation is a physically plausible motion. Like a person opens a mouth, car moves, while interpolation is just warping one image into the other. Fig 7 shows exactly this with the end states being plausible states of the system. The authors should fix the ambiguity to avoid misunderstanding. The authors also don't report any generation results. Can I sample a random shape from the learnt distribution? If not the I don't think it's correct to say the model is generative. 2. Motivation. It's not clear why the problem is important from practical standpoint? Why one would like to interpolate between two icons? Motivation behind animation is more clear, but in my opinion, the paper doesn't do animation. I believe from a practical standpoint letting the user to input text and be able to generate an icon would also be important. Again, I have hard time understanding why shape autoencoding and interpolation is interesting. 3. Experiments. Probably the biggest concern with the paper is with the experiments. The paper reports only self comparisons. The paper also doesn't explain why this is so, which adds to the poor motivation problem. In a generative setting comparisons with SketchRNN could be performed.

Correctness: Given the motivation and terminology issues I feel the authors should correct the paper to remove potential misunderstanding.

Clarity: I think the paper is written very well.

Relation to Prior Work: I think it is discussed sufficiently well. Comparisons with the related work are missing.

Reproducibility: Yes

Additional Feedback: I think it's a good paper and overall I'm positive. The paper presents an interesting method. There are some issues with the paper. It also true that they try to tackle a somewhat new area. I, however, think that in the present form the paper is slightly below the bar as discussed in the weaknesses section, explaining my rating.


Review 3

Summary and Contributions: The paper proposes a new method called DeepSVG that tackles the ambitious task of generation and interpolation of complex vector graphics. It does so by first concentrating on a dataset the authors collected from 'icons8', and design a variational autoencoder-based architecture that handles the data hierarchically. The sequential mechanism is based on Transformers. The generative network first encodes every path (shape) independently, indexed by E1, and combined with E2 into the final representation (bottleneck) which is then similarly decoded back to the input. Finally, a matching between the input & output is done through Ordered Assignment which was found in an ablation study (human ranking) to give good results. Once reconstruction is achieved, interpolation can be done at the latent space. The authors provide very nice figures in the paper, but even more in the supplementary material, including an interactive tool that let the viewer choose the interpolation between two shapes. The authors will release the collected dataset ('SVG-Icons8'), which is to be appreciated as well. I would encourage the authors also the release the code of their method to allow future work to be able to compare to, since the architecture would be hard to genuinely re-implement.

Strengths: 1. Instead of handling the SVG language as a standard language, the paper proposes a thoughtful way of embedding SVG draw-commands, converting tokens into higher dimensional representation through trainable weights. 2. The method is not autoregressive hence more accessible than alternative methods. 3. Achieves better reconstructions and smoother interpolations than alternative baselines based on the ablation human ranking study.

Weaknesses: 1. Measure of success is limited only to human raters (Table 2). This is a challenge with generative models but I would have liked to know more about how well the structures are reconstructed, perhaps rasterized-based measures could be used. 2. Clearly, scalable vector graphics can be extremely complicated (beyond icon8 dataset) so it's unclear how well the method would perform in other datasets, either more complicated ones or out of domain generalization. What happens when the number of paths exceeds X -- in that case would E2 performs favorably?

Correctness: Ablation study is provided.

Clarity: The paper is well written. The method is described in a clear way. The figures are very descriptive, and the supplementary part has many more examples.

Relation to Prior Work: Sufficient but not extensive.

Reproducibility: No

Additional Feedback:


Review 4

Summary and Contributions: The paper propose a hierarchical generative network architecture, which learns to generate scalable vector graphics. The paper also introduce a dataset of SVG icons, and it shows some successful interpolation and manipulation result with the learned generative model.

Strengths: Directly working on rendering commands of svg icons instead of rasterized images is a more complicated problem which the paper shows a sophisticated solution. The general idea of having a hierarchical VAE encoding/decoding a set of commands in SVG is intuitionly a reasonable solution, which is backed by the experiments.

Weaknesses: A few confusions I had for the paper: 1. The generative model is said to be hierarchical, but the factorization shown in Eq.2 is a single layer model with the variables being independent to each other conditioned on the z. 2. The description of the baseline in sec.4.1 is not abundantly clear, the author should either give a detailed description in the paper/supp, or refer to some reference if the model is borrowed. 3. I don't really see noticable difference between a one-stage model v.s. the hierarchical one in Fig.5. The subjective result in Tab.5 is not enough to convince me the quality difference between the two. 4. also, although it is clear to have the decoder output rendering commands, it is not clear why the encoder should directly take input from rendering commands instead of rasterized images, as in [10]. The paper misses comparison in this regard, which a fair comparison against [10] is needed. 5. No reconstruction error on hold out set is reported, therefore we have no idea about the generalization ability of the trained AE.

Correctness: The description of the method is mostly correct, except Eq.2. which I'd like the author to comment. Additional experiment (see above) is in need.

Clarity: The paper is well written.

Relation to Prior Work: It has a clear discussion of related works.

Reproducibility: Yes

Additional Feedback:

[Author Response · NeurIPS 2020]

We thank the reviewers for their thoughtful comments and insights. We are encouraged that reviewers found our work to be tackling a significantly novel (**R1**) task, that is both ambitious (**R3**) and more complicated (**R4**) than its rasterized counterpart. We are glad they found our method to be novel (**R1**), thoughtful (**R3**) and intuitive (**R4**). The release of our collected dataset is appreciated by the reviewers (**R3**) and qualified as interesting for the research community (**R2**). We are also pleased that reviewers liked the figures and interactive tool provided in the supp. material of our work (**R3**). **Code and dataset release:** Since our submission, we have indeed released all code and the Icons-8 dataset.

**R1-W1. More complex SVGs:** Our work is intended as a step towards the ability to process more and more complex vector graphics. Note that the icons data considered in this paper represents a significant increase in complexity compared to the sketch and font datasets used in prior work. Moreover, unlike [4, 10], our model learns representations of *arbitrary* icons, without being conditioned on auxiliary class labels (e.g. glyph unicode or sketch category).

**R1-W2. Readability:** We thank **R1** for the suggestions, and will revise the specified parts to increase readability.

**R1-C1, R4-W4. SVG-VAE:** We aim to design a transformer-based architecture that avoids the extra step of rasterization in the encoder/decoder pipeline. In this regard, SVG-VAE is not our baseline; in fact, our experiments on the fonts dataset (see Sec. F in the supp.) suggest that our baseline performs better than SVG-VAE.

**R1-C2,C3, R4-W2. Baseline:** We regret that the description of our baseline (L192-197) is short due to space limitations. In the final version, we will provide a detailed description of all methods in Tab. 2. As described in Sec. 4.1, our baseline predicts commands autoregressively, while the 'one-stage feed-forward' does it in one forward pass (L127-138).

**R2-W1a. Terminology:** We regard *animation* to be a task and investigate *interpolation* as *one* approach to perform it. We will clarify this further in the revised paper. We believe our generative model could be used as an interactive tool to assist 2D animators in shape morphing between two keyframes. This process can be repeated iteratively – adding a hand-drawn keyframe at every step – until a satisfying result is achieved. **R2-W1b. Generation results:** We provide generative examples, sampled from the learnt distribution of both fonts and icons in Sec. F and G of our supp. material.

**R2-W2. Motivation:** A major purpose of our work is, as for prior VAE designs [10], to learn deep *representations* of SVGs. As also mentioned by **R1**, we believe that this ability has many applications, including image vectorisation, style transfer (investigated by the *squarify* op. in Sec. 4.3), classification, animations, or indeed text-conditional generation.

**R2-W3. Experiments:** Since SketchRNN is only able to process polyline drawings, the only possible direct comparison with prior work is SVG-VAE [10], which we present in Sec. F of the supp. material. We will move it to the main paper.

**R3-W1, R4-W5. Quantitative measures:** We are unaware of quantitative metrics for vector image generation. We therefore here propose two metrics. We first define the Chamfer distance between two SVGs: $d_{\mathrm{Chfr}}(V, \hat{V}) = \frac{1}{N_P} \sum_{i=1}^{N_P} \min_j \int_t \min_\tau \|P_i(t) - \hat{P}_j(\tau)\|_2 dt$, where $P_i \in V$ is a path (L91). The *Reconstruction Error* (RE) is $d_{\mathrm{Chfr}}(V, \hat{V})$ where $V$ and $\hat{V}$ are the GT and reconstruction. The *Interpolation*

Table 1: Metrics on the train / test sets.

|  | RE | IS |
| --- | --- | --- |
| Baseline | 0.10 / 0.17 | 0.25 / 0.36 |
| One-stage feed-forward | **0.007** / 0.014 | 0.12 / 0.17 |
| Ours (Ordered) | **0.007** / **0.012** | **0.08** / **0.12** |

*Smoothness* (IS) is defined as $\sum_{k=1}^{M} d_{\mathrm{Chfr}}(V^{\alpha_{k-1}}, V^{\alpha_k})$, where $M$ is the number of frames, $\alpha_k = k/M$ and $V^\alpha$ is the predicted SVG interpolation parametrized by $\alpha \in [0, 1]$. Results are shown in Tab. 1. Compared the the One-Stage method, our approach achieves improved RE on the test set and significantly better interpolation quality (IS). We will add this experiment along with a discussion and additional details in the final version.

**R3-W2, R4-W5. More complicated SVGs and generalization:** As mentioned in **R1**-W1 above, please recall that the setting we tackle in this work is a significant *increase* in complexity compared to prior works [4,10]. Out of domain representations can be processed and interpolated, as shown in Sec. 4.2, where user-drawn images do not appear in the training set. The maximum number of paths, $N_P$, is simply a parameter that can be adjusted based on the statistics of the dataset. In the extreme case where the number of paths exceeds the specified $N_P$, the SVGs can still be processed by partitioning it into smaller parts, analogous to how CNN-based image generation networks are applied to HD images.

**R4-W1. Eq. 2:** The mentioned equation is intended to clarify that we predict paths and commands in a purely feed-forward manner – as opposed to autoregressive models, whose predictions are conditioned on previous outputs. It covers both the hierarchical and one-stage architectures. In the former setting, as illustrated in Fig. 4, $(\hat{c}_i^j, \hat{X}_i^j))_{j=1}^{N_C} = D^{(1)}(\hat{u}_i)$ is output by $D^{(1)}$, where the path encoding $\hat{u}_i$ is itself predicted as $\{\hat{u}_i\}_{i=1}^{N_P} = D^{(2)}(z)$ based on latent vector $z$. Thus, in the hierarchical case, we can write $p(\hat{X}_i^j | z, \theta) = p(\hat{X}_i^j | \hat{u}_i(z, \theta), \theta)$, and similarly for the other factors in Eq. (2).

**R4-W3. Interpolation quality:** To quantitatively evaluate interpolation quality, we introduce the interpolation smoothness (IS) metric (see **R3**-W1), reported in Tab. 1. The hierarchical model achieves superior performance compared to the one-stage architecture. In particular in the first two examples in Fig. 5 (onion-tree, footstep-shovel), the one-stage model's interpolations suffer from instability, as reflected in the color changes. This results in severe flickering, which is captured by our IS metric (Tab. 1) but is difficult to visualize in printed form. For the human study, the raters were therefore shown looped animations, as included in our interactive supplementary material.

[Meta-Review · NeurIPS 2020]

Three out of four reviewers believe that the paper should be accepted due to the interesting and new application domain, and a reasonable approach. The fourth reviewer is also positive, though she/he believes that the paper could be improved before acceptance. The rebuttal as well as the parts of supmat seem to address part of the concerns of this reviewer, and the authors are encouraged further improve the final version bringing in some results from the supmat into the main text (e.g. comparison with SVG-VAE). On balance the recommendation of the area chair is to accept.